# ThermoMaze behavioral paradigm for assessing immobility-related brain events in rodents

**Mihály Vöröslakos[1†], Yunchang Zhang[1,2†], Kathryn McClain[1], Roman Huszár[1], Aryeh Rothstein[1], György Buzsáki[1,3]***

[1]Neuroscience Institute, New York University, New York, United States; [2]Princeton Neuroscience Institute, Princeton University, Princeton, United States; [3]Department of Neurology, School of Medicine, New York University, New York, United States

---

## eLife assessment

The ThermoMaze represents a **valuable** tool to control the rest/exploration states of an animal. The data, collected and analyzed using **solid** and validated methodology, demonstrate its use in addressing previously elusive questions. This will facilitate future work with more in-depth analysis of place cell activity to further support for some of the claims.

---

**\*For correspondence:**
gyorgy.buzsaki@nyulangone.org

[†]These authors contributed equally to this work

**Abstract** Brain states fluctuate between exploratory and consummatory phases of behavior. These state changes affect both internal computation and the organism's responses to sensory inputs. Understanding neuronal mechanisms supporting exploratory and consummatory states and their switching requires experimental control of behavioral shifts and collecting sufficient amounts of brain data. To achieve this goal, we developed the ThermoMaze, which exploits the animal's natural warmth-seeking homeostatic behavior. By decreasing the floor temperature and selectively heating unmarked areas, we observed that mice avoided the aversive state by exploring the maze and finding the warm spot. In its design, the ThermoMaze is analogous to the widely used water maze but without the inconvenience of a wet environment and, therefore, allows the collection of physiological data in many trials. We combined the ThermoMaze with electrophysiology recording, and report that spiking activity of hippocampal CA1 neurons during sharp-wave ripple events encode the position of mice. Thus, place-specific firing is not confined to locomotion and associated theta oscillations but persist during waking immobility and sleep at the same location. The ThermoMaze will allow for detailed studies of brain correlates of immobility, preparatory–consummatory transitions, and open new options for studying behavior-mediated temperature homeostasis.

---

## Introduction

All behaviors can be considered as parts of a sequence of action–rest transition (*Buzsáki and Tingley, 2023*). Brain states in vertebrates fall into dichotomous categories, and correspond roughly to what early behavioral research referred to as 'preparatory' (or 'exploratory') and 'consummatory' (or 'terminal') classes (*Greaves and Barnett, 1977*). In mammals, these two fundamental brain states can be readily identified by basic electrophysiological monitoring of various brain structures (*Vanderwolf, 1969*). They are also referred to as voluntary and non-voluntary or conscious and non-conscious brain states (*Vanderwolf, 1969*). Switching between these states is correlated with high and low release of subcortical neuromodulators (*Buzsaki et al., 1988*; *Devilbiss and Waterhouse, 2004*; *Dringenberg and Vanderwolf, 1997*; *Harris and Thiele, 2011*; *Metherate et al., 1992*; *McCormick et al., 2020*).

Consummatory/terminal behaviors include feeding and drinking, resting and its extreme form, non-rapid eye movement (NREM) sleep, while preparatory/exploratory behaviors include locomotion and other movements that are a result of a general tendency to sample environmental stimulus. Preparatory and consummatory behaviors in the hippocampus are associated with theta oscillations and sharp-wave ripples (SPW-Rs), respectively (*Buzsáki et al., 1983*).

Deciphering the physiological underpinnings of these categories and revealing the significance of brain state transitions for cognition requires sufficient sampling of the relevant brain states. This is usually achieved by extended repeated recordings or, when possible, recording large numbers of neurons simultaneously. Prolongation of exploratory behavior can be readily achieved by placing the animal in novel environments, by food or water deprivation or by introducing delays in choice behavior tasks (*Carandini and Churchland, 2013*; *Pisula and Siegel, 2005*). Recently, the honey-comb maze paradigm was introduced to extend the observation periods of exploratory deliberation (*Ormond and O'Keefe, 2022*).

In contrast, the experimental control of consummatory behavioral classes is more difficult. Sleep provides an opportunity for long recordings. Comparison of sleep before and after learning is a standard paradigm to examine experience-induced brain plasticity (*Kay and Frank, 2019*; *Wilson and McNaughton, 1994*). Consummatory brain states associated with eating, drinking, and sex change rapidly with satiety and require prolonged periods of deprivation (*Allen et al., 2019*; *Toth and Gardiner, 2000*; *Hughes et al., 1994*; *Collier and Levitsky, 1967*). Controlling periods of awake immobility is most difficult (*Malvache et al., 2016*; *Foster and Wilson, 2006*; *Kay et al., 2016*), mainly because forced immobilization of the animal is stressful (*McEwen et al., 2012*) and is accompanied by altered physiological states (*Foster et al., 1989*).

Here, we introduce the ThermoMaze, a behavioral paradigm that allows for the collection of large amounts of physiological data while the animal rests at distinct experimenter-controlled locations. In standard laboratory environments (20–24°C) (*Garber et al., 1996*), both housing and data collection take place below the thermoneutral zone of mice (26–34°C) (*Škop et al., 2020*; *Maloney et al., 2014*; *Gordon, 1993*). The ThermoMaze exploits the animal's behavioral thermoregulation mechanisms (*Gaskill et al., 2009*; *Kanosue et al., 1998*) and promotes thermotaxis (i.e., movement in response to environmental temperature) (*Gaskill et al., 2011*). Searching for a warmer environment, social crowding, and nest building are natural behavioral components of heat homeostasis (*Gaskill et al., 2011*; *Gordon et al., 1998*; *Chen et al., 1998*). The ThermoMaze allows the experimenter to guide small rodents to multiple positions in a two-dimensional environment. Decreasing the maze floor temperature induces heat-seeking behavior and after finding a warm spot, the animal stays immobile at that spot for extended periods of time, allowing for recording large amounts of neurophysiological data in immobility-related brain states. We report on both behavioral control and hippocampal electrophysiological correlates of heat-seeking activity to illustrate the versatile utility of the ThermoMaze.

## Results

### Design and construction of the ThermoMaze

The ThermoMaze is designed to guide small rodents to warm spatial locations in a two-dimensional cold environment, consisting of a box (width, length, and height: 20, 20, and 40 cm, respectively) made from an acrylic plexiglass sheet (*Figure 1A*, top). The floor of the maze is constructed from 25 Peltier elements (40 × 40 × 3.6 mm) that are attached to aluminum water-cooling block heatsinks (40 × 40 × 12 mm, $n$ = 25) with heat-conductive epoxy and are insulated from each other by wood epoxy (*Figure 1A*, dashed inset). Each Peltier element is controlled by an electrically operated switch (relay) that opens and closes high-current circuits by receiving transistor–transistor-logic (TTL) signals from outside sources (*Figure 1B*). Peltier elements can be heated individually up to 30°C to provide a 'warm spot' for the animal when other regions of the floor are under cooling (*Figure 1B*, active heating of one Peltier element is shown). The ambient temperature of the maze is controlled by water circulated from the water tank through the water-cooling blocks. We set the floor temperature to either ~25°C (room temperature) or to ~10°C (cooling, *Figure 1*, *Figure 1—figure supplement 1*), but a range of ambient temperatures (5–30°C) could be employed. The water temperature is monitored by a K-type thermocouple placed inside the water tank (*Figure 1A*, bottom). The floor temperature of the

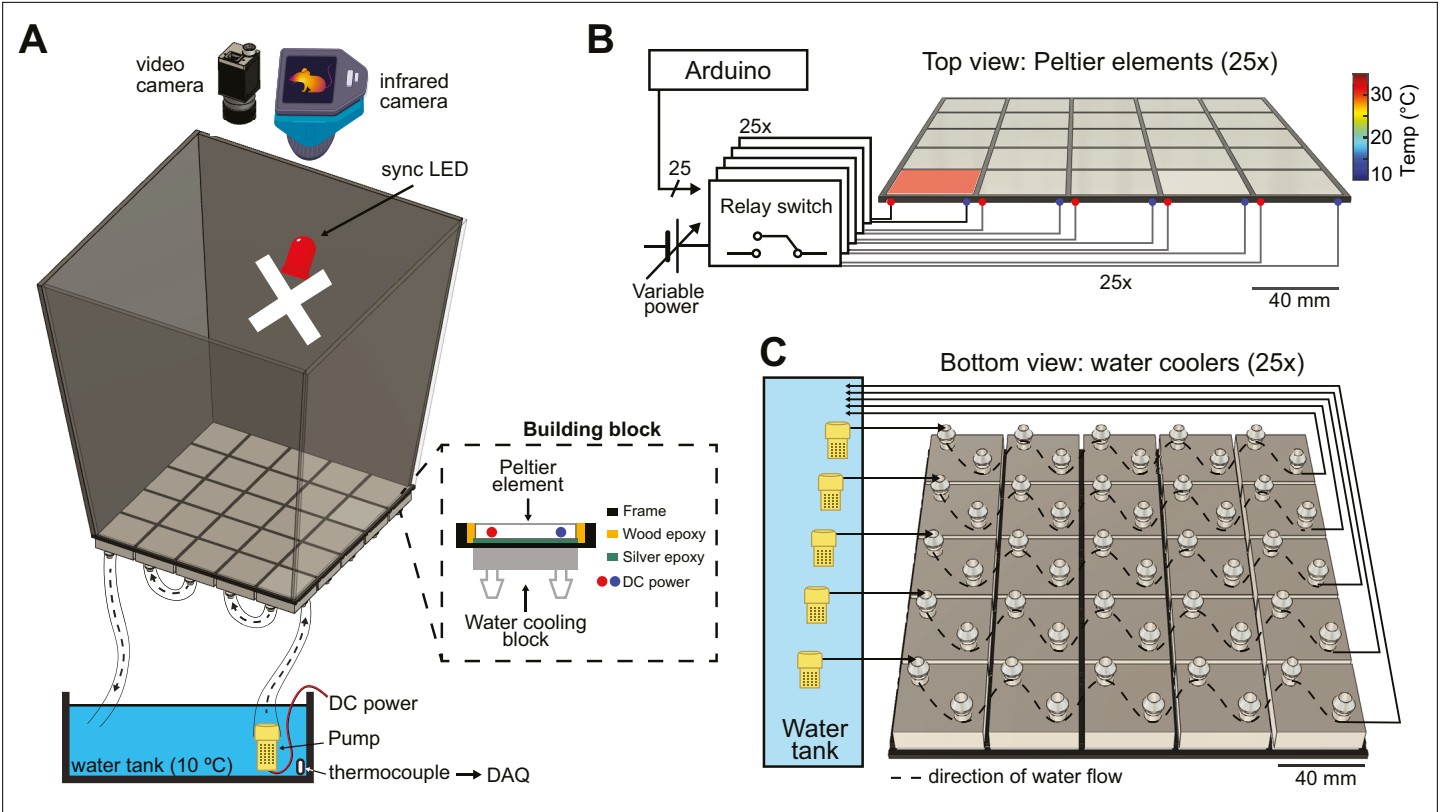

**Figure 1.** Construction and temperature control of the ThermoMaze. (**A**) Schematic of the ThermoMaze. The floor was built using 25 Peltier elements attached to water-cooling block heatsinks (building block). The position of the animal and the temperature of the ThermoMaze can be recorded using a video camera and an infrared camera positioned above the box, respectively. An 'X' was taped inside the maze as an external cue below the camera synchronizing light-emitting diode (LED). Water circulates through the water-cooling heatsinks using a water pump submerged in a water tank (one row of heatsinks is attached to one pump). The temperature of the water tank is monitored and recorded using a thermocouple (white symbol inside water tank, DAQ – analog input of the data acquisition system). Peltier elements are connected to a power supply (red and blue dots represent the anode and cathode connection). (**B**) Circuit diagram and schematic of Peltier elements (*n* = 25), viewed from the top. TTL pulses generated by an AVR-based microcontroller board (Arduino Mega 2560) close a relay switch connected to a variable voltage power source. Each Peltier element can be independently heated (surface temperature depends on applied voltage and temperature difference between hot and cold plate of Peltier element). (**C**) Schematic of the water circulation cooling system, viewed from the bottom of the floor (each Peltier element has its own water-cooling aluminum heatsink, shown in silver, *n* = 25). Five submerging DC pumps are used to circulate water across 25 heatsinks (dashed lines show the Peltier elements connected to one pump). The temperature of the heatsink is transferred to the Peltier element passively through the silver epoxy resulting in passive cooling of the floor of the ThermoMaze.

The online version of this article includes the following figure supplement(s) for figure 1:

**Figure supplement 1.** Control of heating and cooling of the surface of ThermoMaze.

ThermoMaze is monitored using a thermal camera (FLIR C5) providing continuous registration of real-time temperature changes (*Figure 1A*).

Prior to the experiments, the thermal camera, which continuously measures the surface temperature of the floor of the ThermoMaze is calibrated by thermocouples placed directly on Peltier elements (*Figure 2*). The accuracy of the FLIR 5C infrared camera is ±3°C. With proper calibration and attention to emissivity (an object's ability to emit rather than reflect infrared energy) the margin of error can be less than 1°C (*FLIR, 2016*).

## Mice seek out hidden warm spots in the ThermoMaze

To illustrate the novel advantages of the ThermoMaze on behavior and brain activity, we tested 11 mice (*n* = 3 male and 8 female mice) with silicon probe recordings from the hippocampus (*Supplementary file 1a*). One wall was marked by a prominent visual cue (black tape and blinking light-emitting diode) to provide a distinct spatial cue in the box (*Figure 1A*; *Muller and Kubie, 1987*). On

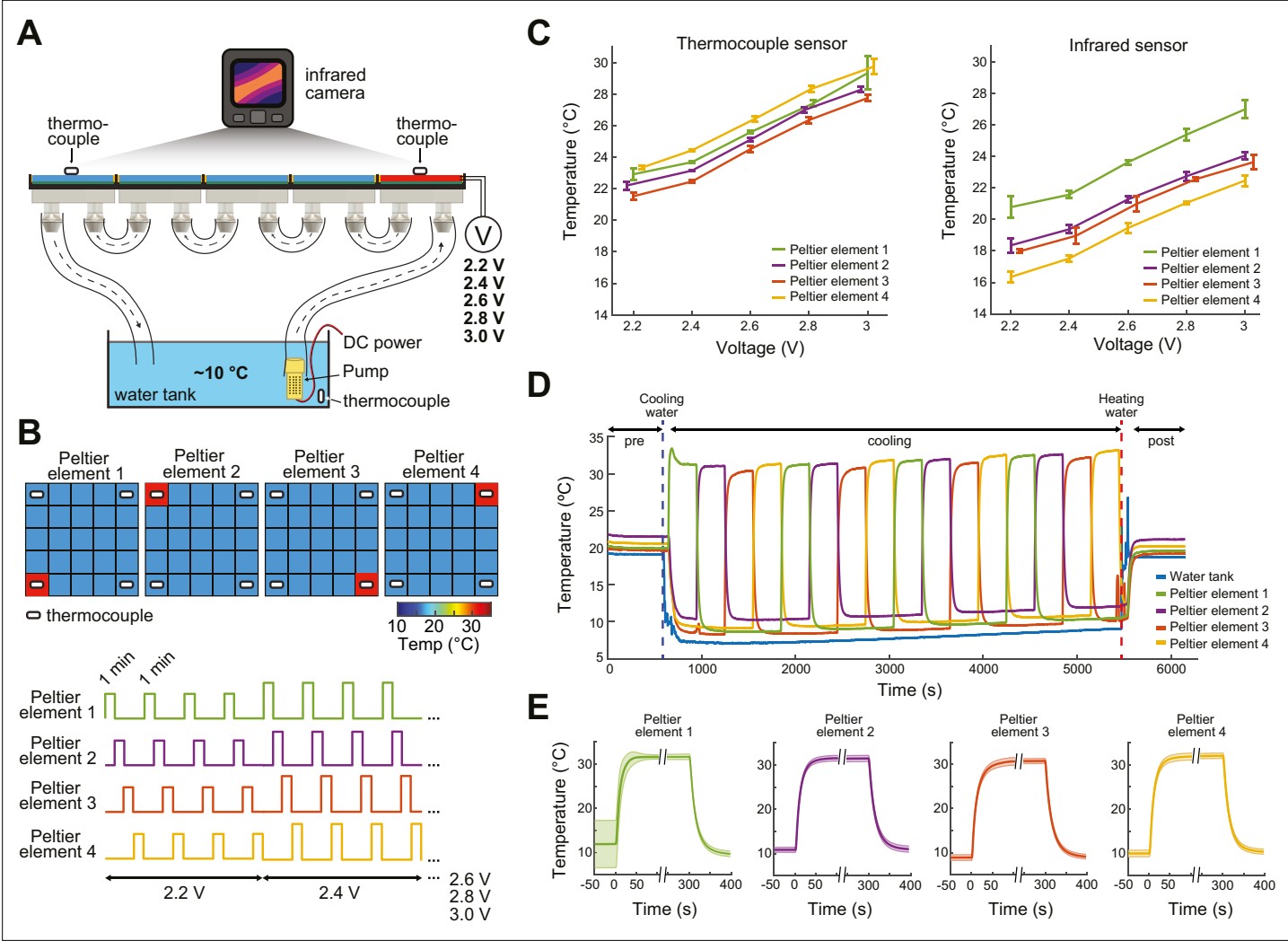

**Figure 2.** Calibration of the ThermoMaze temperature regulation. (**A**) Side view of the ThermoMaze. Prior to animal experiments, we calibrated the heating and cooling performance of the Peltier elements and temperature measurement. We attached thermocouples (white symbols) to the surface of the Peltier elements serving as the ground-truth for calibrating the infrared camera placed above the ThermoMaze. Different voltage levels were used for the calibration (2.2, 2.4, 2.6, 2.8, and 3 V) while the water tank temperature was kept constant. (**B**) Top: four Peltier elements used in later experiments are chosen for calibration (four corners). Bottom: 1-min heating was repeated four times at each voltage level. (**C**) Simultaneously recorded temperature by thermocouples (left) and infrared camera (right). Increasing voltages induced increased heating (*n* = 4 trials per intensity, mean ± SD are shown). While the temporal dynamics yielded similar results between the two systems, we found ~4°C offset between infrared and thermocouple-measured signals. (**D**) Temperature changes of four Peltier elements used during an emulated behavioral session (without any animal subject) tracked by thermocouples. (**E**) Temporal dynamics of temperature changes at the four Peltier elements during active heating and following passive cooling. The temperature reaches steady state within 31 ± 10.3 s (mean ± SD, *n* = 4 trials across 4 Peltier elements).

each experimental day, the mouse was placed in the ThermoMaze and allowed to explore it for 10 min at room temperature ('Pre-cooling' sub-session; *Figure 3A*). Next, the ThermoMaze temperature was decreased to around 14°C for 80 min and four Peltier elements ('warm spots'; typically, in the corners) were sequentially and repeatedly turned on and heated up to 30°C. One Peltier element was turned on for 5 min in a sequential order (1–2–3–4) and the sequence was repeated four times ('Cooling' sub-session; *Figure 3B*). The Cooling sub-session was divided into 5 min 'warm spot epochs' for analysis. The daily experimental session ended with a 'Post-cooling' sub-session (free exploration at room temperature for 10 min). In addition, all mice were recorded in the home cage both before and after the experimental session (*Figure 3A*). During Pre- and Post-cooling sub-sessions, the animal explored the maze relatively evenly with a moderate movement speed (*Figure 3B–D*), although thigmotaxis was the dominant pattern, with corners as highly preferred sites of both movement and immobility

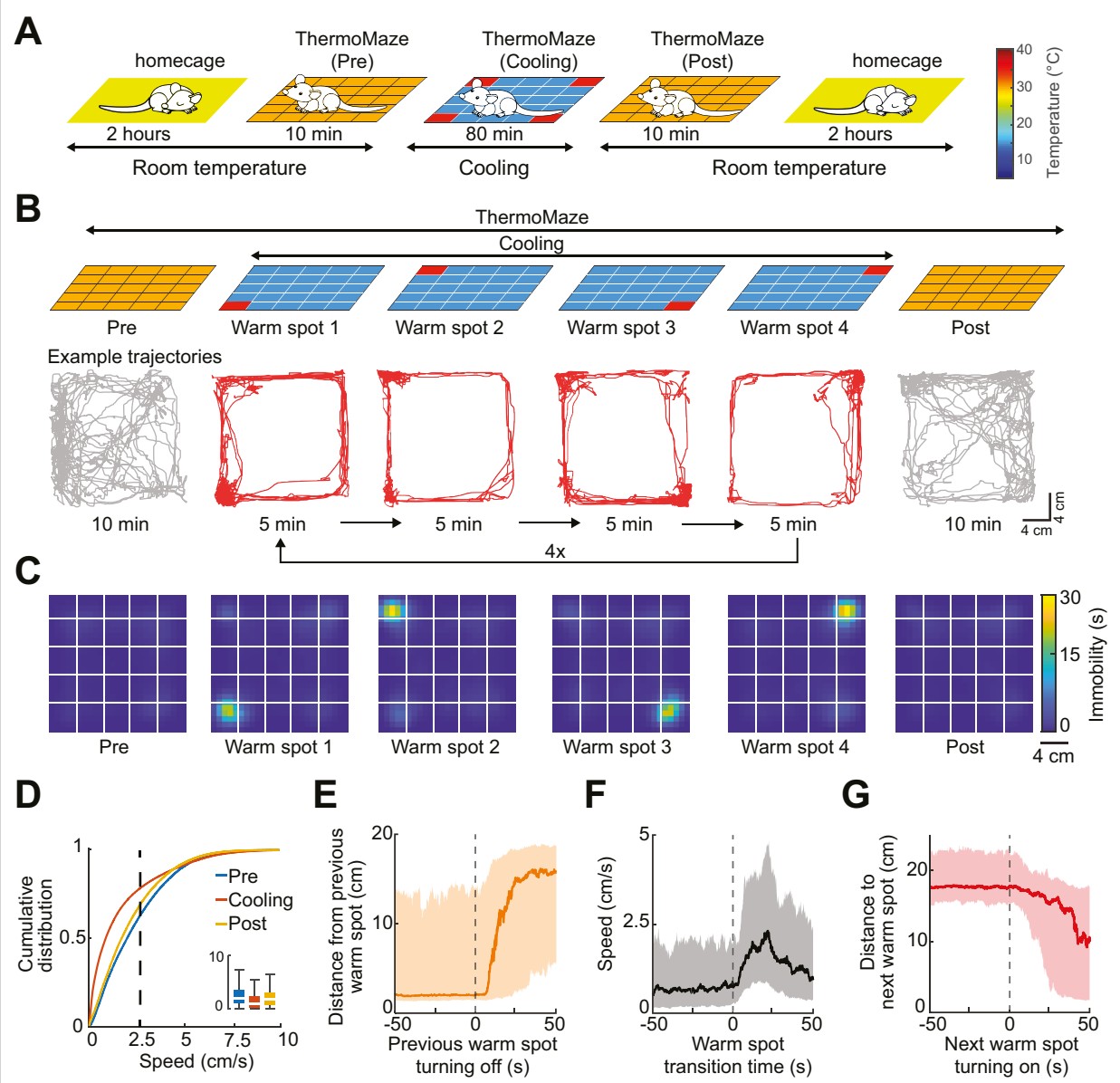

**Figure 3.** Mice track and stay immobile on hidden warm spots in the ThermoMaze. (**A**) Five sub-sessions (epochs) constituted a daily recording session: (1) rest epoch in the home cage, (2) pre-cooling exploration epoch (Pre), (3) Cooling, (4) post-cooling exploration epoch (Post), and (5) another rest in the home cage. (**B**) Schematic of temperature landscape changes when the animal is in the ThermoMaze (top) and example animal trajectory (below). During Cooling, one Peltier element always provided a warm spot for the animal (four Peltier elements in the four corners were used in this experiment). Each Peltier element was turned on for 5 min in a sequential order (1–2–3–4) and the sequence was repeated four times. (**C**) Session-averaged duration of immobility (speed ≤2.5 cm/s) that the animal spent at each location in the ThermoMaze; color code: temporal duration of immobility (s); white lines divide the individual Peltier elements; $n$ = 17 session in 7 mice. (**D**) Cumulative distribution of animal speed in the ThermoMaze during three sub-sessions from seven mice. Median, Kruskal–Wallis test: $H$ = 139304.10, d.f. = 2, p < 0.001. (**E**) Animal's distance from the previously heated Peltier element site. (**F**) Speed of the animal centered around warm spot transitions. (**G**) Animal's distance from the target warm spot as a function of time (red curve: median; time 0 = onset of heating). In all panels, box chart displays the median, the lower and upper quartiles (see **Supplementary file 1b** for exact p-values and multiple comparisons).

The online version of this article includes the following figure supplement(s) for figure 3:

**Figure supplement 1.** Animals learned to track and stay immobile on hidden warm spots in the ThermoMaze.

**Figure supplement 2.** Brain temperature is not affected by cooling of the ThermoMaze.

**Figure supplement 3.** Behavior in the ThermoMaze did not alter hippocampal power spectra.

*Figure 3 continued on next page*

*Figure 3 continued*

**Figure supplement 4.** Spatial distributions of immobility duration and SPW-Rr occurrence are more uniform during Cooling compared to room temperature.

**Figure supplement 5.** Changing the location of warm spots shape behavior.

**Figure supplement 6.** Spatial tuning of hippocampal pyramidal cells in the ThermoMaze.

**Figure supplement 7.** Quantification of spatial tuning properties of CA1 pyramidal neurons as the animal moved through the ThermoMaze.

(*Figure 3—figure supplement 1*). The animals readily found the location of the warm spot after a few training sessions (median = 3). Changing the warm spot locations during Cooling induced locomotion until the mouse found another warm spot and stayed on it for prolonged periods (*Figure 3B, C*, *n* = 17 sessions in 7 mice, *Video 1*). Duration spent on the warm spot roughly followed a bimodal distribution with a median = 2.85 min (*Figure 3—figure supplement 1A*). Compared to Pre and Post sub-sessions, during Cooling, mice spent a smaller proportion of time in locomotion (Pre: 40 ± 19%, Post: 34 ± 16%, Cooling: 23 ± 12%, mean ± SD, defined as speed >2.5 cm/s, *n* = 20 sessions from 7 mice; *Figure 3D*) and more time in immobility (Pre: 59 ± 19%, Post: 66 ± 16%, Cooling: 76.74 ± 12.41%, mean ± SD, defined as speed ≤2.5 cm/s; *n* = 20 sessions from 7 mice; *Figure 3D*). The mice spent most of the time in the corners of the ThermoMaze where heat was provided (*Figure 3—figure supplement 1B*), compared to Pre- and Post-cooling (*Figure 3C*). Once the heating of the Peltier element was turned off, the animal quickly left the warm spot (median duration = 12.99 s, *n* = 20 sessions from 7 mice; *Figure 3E*) and searched for a new source of warmth. Mice increased their speed from 0 to 2.5 cm/s within 12.28 s after a warm spot was turned off (median, *n* = 20 sessions from 7 mice; *Figure 3F*) and found the new warm spot within 23.45 s (median, *n* = 20 sessions from 7 mice; *Figure 3G*). In two additional male mice, we examined brain temperature changes during the Cooling sub-session by implanting a thermistor in the hippocampus (*Figure 3—figure supplement 2A*). In support of previous findings, we found brain state-dependent fluctuation of brain temperature (*Figure 3—figure supplement 2B*; *Petersen et al., 2022*; *Moser et al., 1993*; *Kiyatkin, 2010*). However, cooling the environment per se did not correlate with brain temperature changes (*Figure 3—figure supplement 2C–E*), confirmation that brain temperature is strongly regulated and is largely independent of the ambient temperature (*Kiyatkin, 2010*). The ThermoMaze provides an affordance for mice to select their environmental temperature through the activation of behavioral thermoregulation (*Gordon, 1985*). We also quantified the changes in local field potential (LFP) (1–40 Hz) to test whether the brain state of the animal was similar in the ThermoMaze and home cage during wakeful periods. We did not find significant changes in these frequencies (*Figure 3—figure supplement 3*).

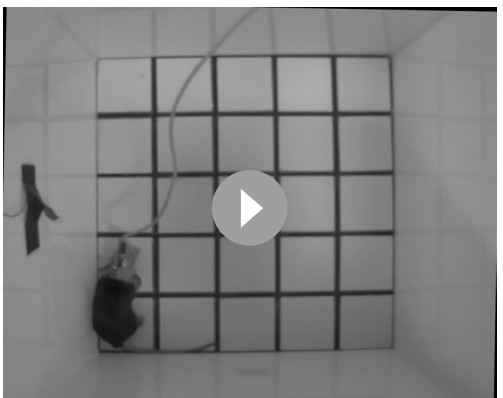

**Video 1.** Real and thermal image of a mouse in the ThermoMaze. The animal's behavior was recorded with a Basler camera and an infrared thermal camera placed above the ThermoMaze. Four Peltier elements were subsequently heated (one in each corner). Infrared image is overlaid on the raw video. The second half of the video is 10 times faster than real time (10× speed legend in the video).

https://elifesciences.org/articles/90347/figures#video1

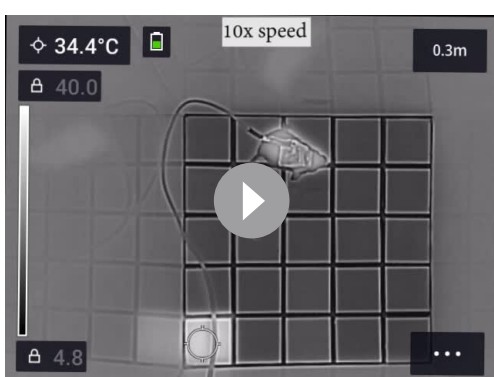

**Video 2.** Thermal image of a mouse in the ThermoMaze. The animal's behavior was recorded with an infrared thermal camera placed above the ThermoMaze (thermal image is in grayscale). In this video, a Peltier element in the inner part of the floor was heated. The speed of the video is 10 times faster than real time.

https://elifesciences.org/articles/90347/figures#video2

One of the objectives in developing the ThermoMaze was to induce immobility at several locations repeatedly and for extended periods. To confirm that this objective was achieved, we ran control sessions with the same duration as the Cooling sub-session but at room temperature (80 min; *Figure 3—figure supplement 4*). Under room temperature conditions (three sessions in three mice), mice first explored the ThermoMaze and settled in one of the corners for an extended period. Although mice spent a similar total amount of time immobile under both conditions, the spatial distribution of immobility durations was more uniform in the Cooling sub-session (*Figure 3—figure supplement 4*) because the ThermoMaze paradigm forced the animals to leave their chosen spot and move to the experimenter-designated locations, that is, the new warm spots away from the corner (*Figure 3—figure supplement 5, Video 2*).

### Firing rate maps of hippocampal neurons in the ThermoMaze

Compared to spatial learning and memory paradigms such as the Morris water maze (*Morris, 1984*), the ThermoMaze has a non-aqueous environment and thus allows for an easy setup of electrophysiological recording. We recorded neurons from the CA1 hippocampal region by multi-shank silicon probes and separated them into putative pyramidal cells and interneurons (Methods – Unit isolation and classification section). We separated behavioral states (movement or immobility) based on movement speed (speed ≧2.5 cm/s = movement and speed <2.5 cm/s = immobility).

To construct spike-count maps for comparing sub-sessions, the ThermoMaze was divided into 25 × 25 bins and the number of spikes emitted by a neuron in each bin was counted and normalized by the time the mouse spent in each spatial bin. The impact of cooling during movement (theta state) was compared by calculating the correlation coefficients between Pre and Post, Pre and Cooling, and Cooling and Post spike-count maps (*Figure 3—figure supplement 6A*). The correlation coefficients decreased significantly across all sub-sessions, with the largest change observed between Pre- and Post-cooling spike-count maps in the experimental mice (*Figure 3—figure supplement 6B*). Thus, the Cooling sub-session in the ThermoMaze induced a moderate decorrelation of pyramidal cells' rate maps, potentially explained by place field remapping upon changing environmental sensory cues (here, temperature gradient) (*Leutgeb et al., 2005*). Such observation constrained our ability to decode spatial information from the spiking activity during SWP-Rs in the Cooling sub-session using firing rate maps constructed during Pre- and Post-cooling sub-sessions (*Zhang et al., 1998*), because the Bayesian decoding approaches have an underlying assumption that the spatial representation (tuning functions, or rate maps) is temporally stable.

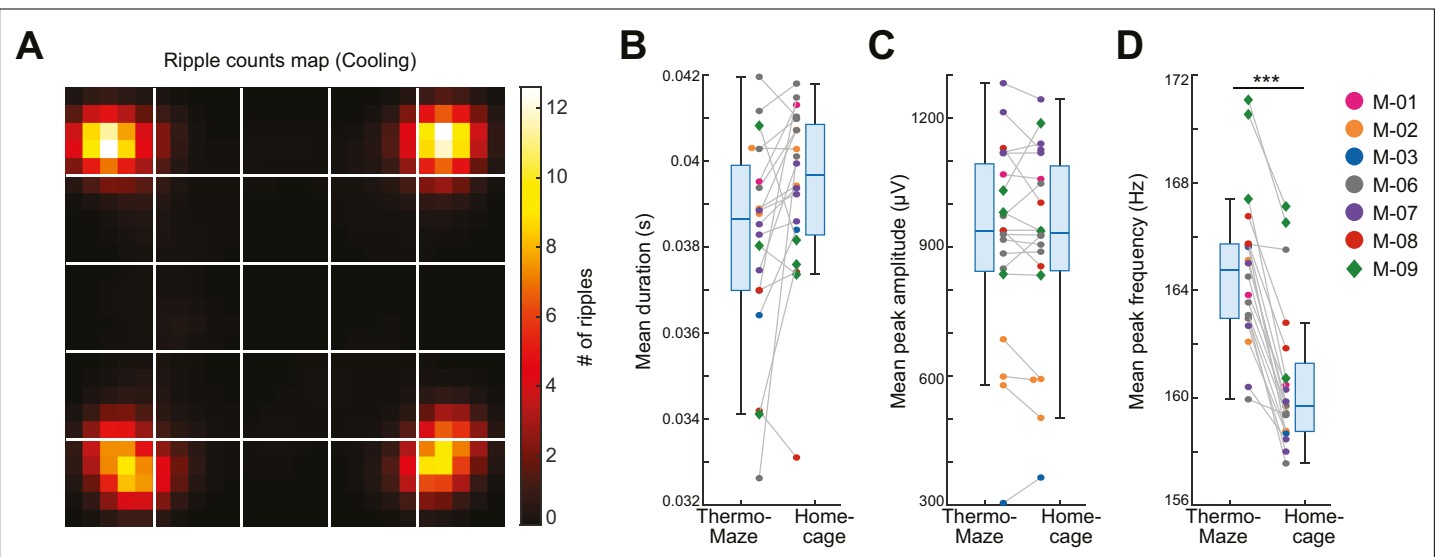

**Figure 4.** Location-specific distribution of SPW-R in the ThermoMaze. (**A**) Spatial map of the number of SPW-Rs during the Cooling sub-session averaged across all sessions (color code: average number of SPW-Rs per session at each location). Session-average number of SPW-Rs during Cooling was 627.3 (corresponding to 0.136 Hz). (**B–D**) Box plots of SPW-R properties in ThermoMaze and in the home cage (*n* = 19 sessions in *n* = 7 mice). (**B**) Mean ripple duration in seconds (s; p = 0.108). (**C**) Mean ripple amplitude in µV (p = 0.9). (**D**) Mean ripple peak frequency in Hz (p < 0.001). Dots (females) and diamonds (males) of the same color represent the same animal. *p < 0.05, **p < 0.01, ***p < 0.001.

Because the ThermoMaze is a relatively small enclosure, one possibility is that CA1 neurons encode less spatial information and only a small number of place cells could be found. Therefore, we identified place cells in each sub-session separately (see more details in the Method section for place cell definition, *Figure 3—figure supplement 7*). We found 40.90%, 45.32%, and 41.26% of pyramidal cells to have place fields in the Pre-cooling, Cooling, and Post-cooling sub-sessions, respectively. Furthermore, we found that, on average, 17.36% of pyramidal neurons passed the place field criteria in all three sub-sessions in a daily session. Therefore, the decorrelation of spatial firing maps across sub-sessions cannot be explained by poor recording quality or weak neuronal encoding of spatial information but is potentially due to changes in environmental cues.

In principle, comparison of place maps during the first and last 10 min of a 100-min session at room temperature should serve as controls. However, at room temperature mice 'designate' one of the corners as home base after a few minutes of exploration and stay in that corner for the rest of the session (*Figure 3—figure supplement 4A*). Thus, exploration of the maze at the end of the session was not available.

## Place-selective neuronal firing during SPW-Rs at experimenter-designated locations

As expected, SPW-Rs occurred predominantly in the corners (*Figure 4A*), where the mice spent most of their time resting (*Figure 3C*). Compared to room temperature control sessions where animals spent most of their time in one corner, the spatial distribution of SPW-Rs in the Cooling sub-session was more uniform (*Figure 3—figure supplement 4A–D*), indicating that the ThermoMaze paradigm successfully biased where SPW-Rs were generated. The duration and amplitude of SPW-Rs were comparable in the ThermoMaze and the home cage (*Figure 4B, C*), whereas the mean peak frequency of SPW-Rs was significantly lower (*Figure 4D*). This decrease can be explained by the lower brain temperate during sleep, a state in which the animals spent most of their time in the home cage (*Petersen et al., 2022*).

To quantify spatial tuning features of neuronal firing during SPW-Rs in the ThermoMaze during the Cooling sub-session, we defined a metric referred to as 'spatial tuning score' (STS). We first binned the floor of the ThermoMaze into four quadrants (2 × 2). For each neuron, we calculated its average firing rate within SPW-Rs in each quadrant. STS was then defined by the firing rate in the quadrant with the highest within-SPW-R firing rate divided by the sum of the within-SPW-R firing rates in all four quadrants (yielding a value between 0 and 1; *Figure 5A*). To test the significance of STS, we compared the STS values with their shuffled versions by randomly assigning one of the four quadrants to each SPW-R. The distribution of the STS in actual SPW-Rs was significantly higher compared to shuffled controls (*Figure 5B*). Additionally, pyramidal cells exhibited higher STSs compared to interneurons (medians: pyramidal cells = 0.3432; interneurons = 0.2934; one-sided Wilcoxon rank sum test, p < 0.001). In summary, both excitatory and inhibitory neuronal populations exhibit place-selective firing during SPW-Rs, while the excitatory neurons demonstrate a stronger place-specific firing.

To quantify how well CA1 neurons encode spatial information during SPW-Rs at the population level, we carried out a Bayesian decoding analysis to read out the current position of the animal from spiking activity (*Zhang et al., 1998*). We constructed firing rate map templates using spikes within SPW-Rs in the training dataset and determined animal positions that maximized the likelihood of observing the spike train during SPW-Rs in the testing dataset (see Method). Spiking activity during SPW-Rs reliably identified the quadrant that the animal was in above chance level (*Figure 5C, D*) irrespective of whether we incorporated the spatial distribution priors into the decoder in an example session (*Figure 5C*) or used a uniform prior (*Figure 5D*).

To relate the spatial content of spikes during SPW-Rs and locomotion, we examined whether the same or different groups of neurons contributed to the place-specific firing during SPW-R and locomotion by calculating the firing rate ratios within the preferred quadrant versus all quadrants. These ratios during SPW-Rs and movement were positively correlated (*Figure 5E*; n = 1150 pyramidal cells in 20 sessions from 7 mice), suggesting that place cells (*O'Keefe and Nadel, 1978*) during movement preserved their spatial properties during SPW-Rs (see also *Figure 5—figure supplement 1* for further analysis and findings on interneurons).

Finally, we tested whether the preservation of spatial features of neuronal spiking also holds at the population level by constructing population vectors separately during movement and SPW-Rs. We

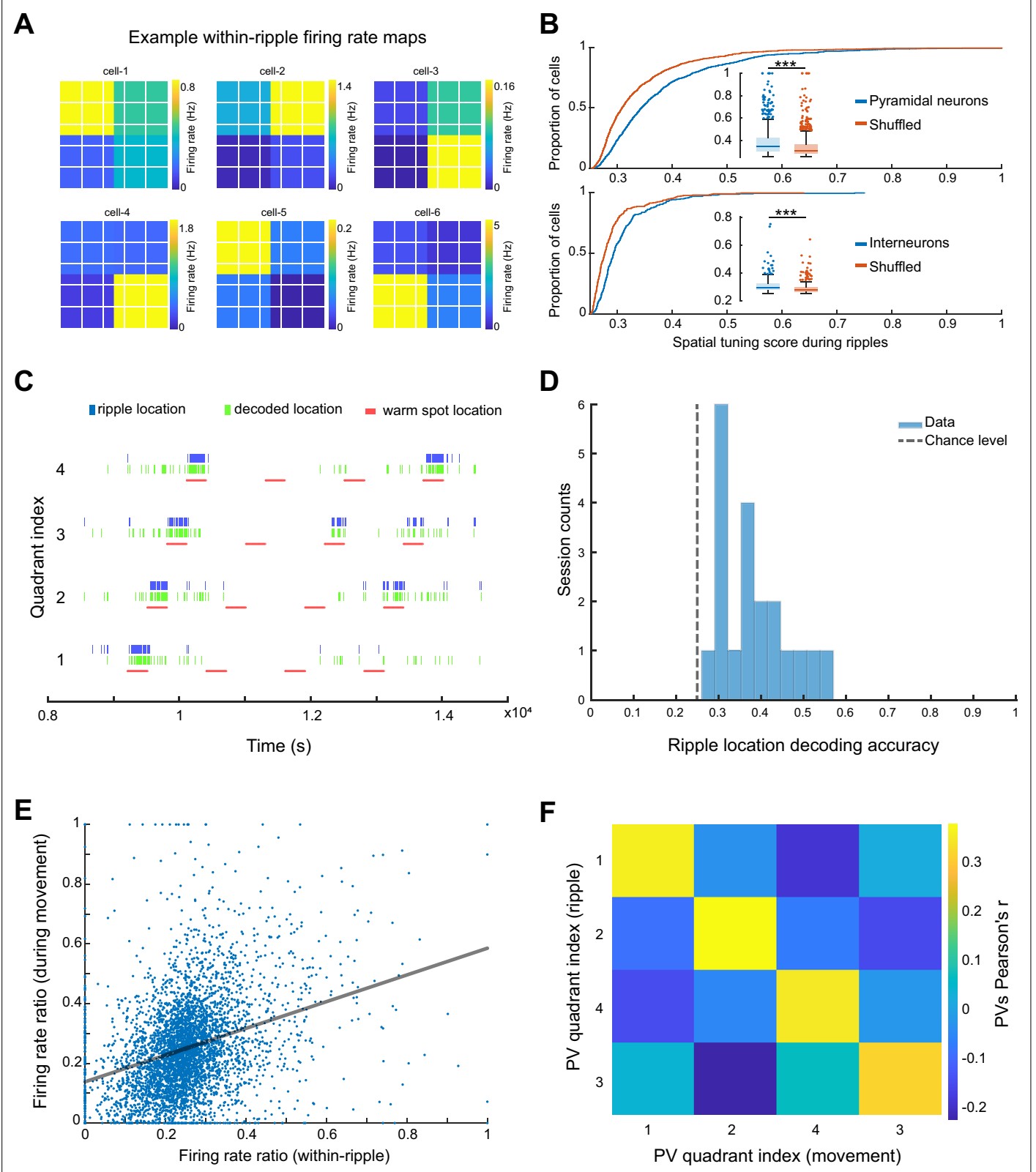

**Figure 5.** Spikes of CA1 pyramidal neurons during awake SPW-Rs are spatially tuned. (**A**) Within-SPW-R firing rate maps (ThermoMaze divided into four quadrants) of six example cells with high within-SPW-R spatial tuning score (STS; from left to right, top to bottom, STS = 0.458, 0.639, 0.592, 0.672, 0.655, and 0.660, respectively). Color represents within-SPW-R firing rate (in Hz) of the neuron in each quadrant of the ThermoMaze. (**B**) Cumulative distribution of STSs of pyramidal neurons (top; *n* = 1150; p < 0.001) and interneurons (bottom; *n* = 288; p < 0.001) during SPW-Rs. Chance levels were calculated

*Figure 5 continued on next page*

*Figure 5 continued*

by shuffling the quadrant identity of the SPW-Rs. One-sided Wilcoxon rank sum tests. (**C**) Bayesian decoding of the mouse's location (quadrant of the ThermoMaze) from spike content of SPW-Rs in an example session (blue: actual ripple location; green: decoded locations; red: locations of the warm spot; session decoding accuracy = 0.65; chance level = 0.26). (**D**) Histogram of session Bayesian decoding accuracies of ripple locations using spiking rate maps constructed during ripples as templates (with a uniform prior and a 100-fold cross-validation; p < 0.001). One-sample *t*-test. (**E**) Firing rate ratios of pyramidal cells constructed during SPW-Rs and movement are positively correlated (Pearson's *r* = 0.321, p < 0.001). The firing rate ratio measures the firing rate of a cell in one quadrant versus the sum of its firing rates in all four quadrants under a specific condition (within-ripple or during movement). (**F**) Matrix of the pairwise correlation coefficient between each pair of firing rate ratio population vectors constructed during SPW-Rs and movements in different quadrants (*x* and *y* axes). Color represents Pearson's *r*. *p < 0.05, **p < 0.01, ***p < 0.001.

The online version of this article includes the following figure supplement(s) for figure 5:

**Figure supplement 1.** Comparison of firing patterns of pyramidal cells and interneurons during SPW-Rs.

then computed the pairwise correlation coefficients between these two conditions. As was the case for individual pyramidal cells, population vectors for the same quadrant during movement were similar to those during SPW-Rs (*Figure 5F*). Overall, these findings support and extend the observation that spiking activity during SPW-Rs continues to be influenced by the animal's current position (*O'Neill et al., 2006*).

To test specifically whether perceptual sensing of environmental features is critical in position-specific firing of neurons during SPW-Rs, we prolonged the duration of warm spots. After the Pre-cooling sub-session, the ThermoMaze temperature was decreased to 16°C for 80 min and two Peltier elements were heated alternately to 30°C for 20 min (*Figure 6A*). As expected, mice spent most of the time immobile on the warm spots (*Figure 6A, B*). Similar to the 5-min protocol (*Figure 4A*), SPW-Rs occurred predominantly on the warm spots (*Figure 6C*). The increased duration of stay on the warm spot facilitated the occurrence of sleep, as quantified by our brain state scoring algorithm (*Figure 6D*, SPW-Rs). REM sleep was not detected since REM state typically emerges after 20–30 min of NREM episodes (*Watson et al., 2016*). Mice spent a higher fraction of their time in sleep during the 20 min, compared to the 5 min sub-session (p = 0.003, *n* = 19 sessions in 7 mice and *n* = 7 sessions in 4 mice, *Supplementary file 1a*). The average inter-NREM interval was 1000s (*Figure 6F*, *n* = 7 sessions in 4 mice). Comparing the population vector similarity of waking SPW-Rs versus NREM SPW-Rs to movement versus waking SPW-Rs and movement versus NREM SPW-Rs, we found that place-specific coding during SPW-Rs persists into sleep, and we observed the highest correlation when comparing SPW-Rs during different brain states than when comparing SPW-Rs and movement (*Figure 6G*). These findings further support the view that sensory inputs during waking SPW-Rs can affect the spiking content of SPW-Rs.

Since sleep occurred on the warm spots during the prolonged stays, we also tested our hypothesis that the difference in the mean ripple peak frequency (*Figure 4D*) between the home cage and ThermoMaze was due to the sleep versus non-sleep states. We compared the ripple peak frequency that occurred during wakefulness and NREM epochs in the home cage and ThermoMaze (*n* = 7 sessions in 4 mice). We found that the peak frequency of the awake ripples was higher compared to both home cage and ThermoMaze NREM sleep (one-way ANOVA with Tukey's posthoc test; ripple frequencies were: 171.63 ± 11.69, 172.21 ± 11.86, 168.19 ± 11.10, and 168.26 ± 11.08 Hz, mean ± SD for home cage awake, ThermoMaze awake, home cage NREM, and ThermoMaze NREM conditions, p < 0.001 between awake and NREM states).

## Discussion

To investigate the importance of brain state transitions in a controlled manner, we developed the ThermoMaze, a behavioral paradigm that allows for the collection of large amounts of physiological data while the animal rests at distinct experimenter-controlled locations. Since the paradigm exploits natural behavior, no extensive training or handling is necessary. We demonstrate that mice regularly explore a cold environment until a warm spot is identified. They spend most of the time in a warm spot and even fall asleep, thus exhibiting a high degree of comfort. We exploited the long immobility epochs following exploration and showed how neurons active during hippocampal SPW-Rs replay waking experience. The ThermoMaze will allow for detailed studies of brain correlates of preparatory–consummatory transitions and open new options for studying temperature homeostasis.

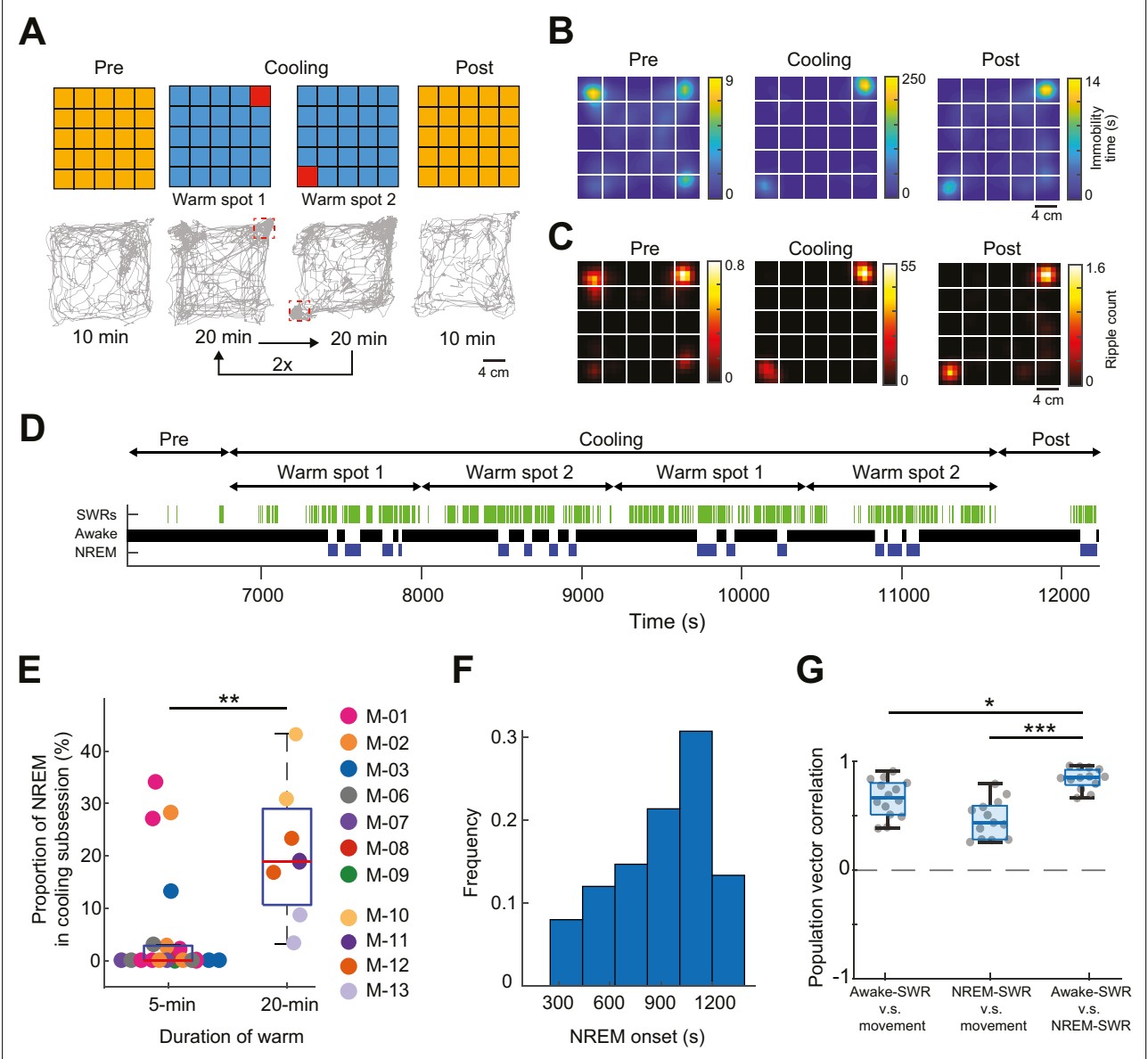

**Figure 6.** Mice sleep at experimenter-defined locations. (**A**) Schematic of ThermoMaze with warm spot locations (top) and the trajectory of an example animal (bottom; red rectangles correspond to the location of warm spots). During Cooling, one Peltier element was turned on for 20 min followed by another (1–2) and the sequence was repeated two times. (**B**) Session-averaged duration of immobility (speed ≤2.5 cm/s) at each location in the ThermoMaze; white lines divide the individual Peltier elements ($n$ = 7 sessions, $n$ = 4 mice). (**C**) Spatial distribution of SPW-R occurrences (color code: average number of SPW-Rs per session at each location, $n$ = 7 sessions, $n$ = 4 mice). Session-average of SPW-Rs during Cooling was 775 (corresponding to 0.16 Hz). (**D**) Long duration of heating allowed for non-rapid eye movement (NREM) sleep occurrence during Cooling sub-session in an example session. Brain state changes (*Watson et al., 2016*) are shown together with SPW-Rs (green ticks). Note that NREM sleep occurs in the second half of the 20-min warming. (**E**) Mice spent a larger fraction of time in NREM during 20-min Cooling sub-session compared to the 5-min task variant (**$p$ = 0.003, $n$ = 19 sessions in 7 mice and $n$ = 7 sessions in 4 mice). (**F**). Mice typically spent ~1000 s awake between NREM epochs. (**G**) Box charts of Pearson's correlation coefficients between population vectors of CA1 pyramidal neurons constructed during awake SPW-Rs, movement, and NREM SPW-Rs. Median, Kruskal–Wallis test: $H$ = 20.7, d.f. = 2, $p$ < 0.001 (pairwise comparison: *$p$ = 0.037 and ***$p$ = 1.6 × 10$^{-05}$).

## Warmth-seeking homeostatic behavior

There is a renewed interest in exploiting natural learning patterns, as opposed to training animals for performing complex arbitrary signal–action associations (*Breland and Breland, 1961*; *Buzsáki, 1982*; *Seligman, 1970*; *Boakes et al., 1978*; *Green et al., 2022*; *Krakauer et al., 2017*; *Brette, 2019*; *Cisek, 2019*). In poikilotherm animals (species whose internal temperature varies with environmental

temperature), energy homeostasis is one of the most fundamental homeostatic processes. Heat homeostasis involves multiple levels of coordination from cellular to systems, from peripheral to central (*Morrison and Nakamura, 2019*; *Tansey and Johnson, 2015*). To maintain core body temperature, thermogenic tissues rapidly increase glucose utilization by brown adipose tissue and shivering by skeletal muscle (*Vallerand et al., 1987*; *Maickel et al., 1967*). The hypothalamic preoptic area (POA) is regarded as the most important thermoregulatory 'center' in the brain (*Harding et al., 2018*; *Song et al., 2016*). Connecting this area of research to learning, the POA is bidirectionally connected with the limbic system and multiple cortical areas which assist both online maintenance of body temperature and preparing the body for future expected changes (allostasis) (*McEwen et al., 2012*; *Sterling and Eyer, 1988*; *Sterling, 2004*). These allostatic mechanisms induce exploratory behavior, searching for a warmer environment (*Schulkin and Sterling, 2019*; *Tan and Knight, 2018*). A location that provides a warm shelter needs to be remembered and generalized for future strategies. Our paradigm offers means to investigate exploratory–consummatory transitions, wake–sleep continuity in the same physical location and, in the reverse direction, the physiological processes that evaluate discomfort levels, motivate behavioral transition from rest to exploration, and the circuit mechanisms that give rise to overt behaviors.

Mice, and rodents in general, are acrophobic and agoraphobic and tend to avoid open areas. Instead, they tend to move close to the wall and spend most of their non-exploration time in corners (*Steimer, 2011*). Thus, while we were able to train mice to seek out and stay in warm spots in the center of the maze after extensive training, their evolutionary 'counter-preparedness' (*Seligman, 1970*) to stay in predator-prone open areas competed with the reward of warming. While these trained mice did stay transiently in the central warm spot, they spent more time returning to the corners. Our mice were on a normal day–light schedule thus their training during the day coincided with their sleep cycle. This explains why after 5–10 min spent on the safe and temperature-comfortable corner warm spots they regularly fell asleep. Yet, we noticed that mice did not simply transition from walking to immobility but, instead, even after finding the warm spot they regularly and repeatedly explored the rest of the maze before returning to the newly identified home base. By changing the temperature difference between the environment and the warm spot, it will be possible to generate psychophysical curves to quantify the competition between homeostatic and exploratory drives in future experiments. These measures, in turn, could be used to study the impact of perturbing peripheral and central energy-regulating mechanisms.

For several applications, there is no need to tile the entire floor of the maze with Peltier elements. For example, a radial-arm maze with cooled floors or placed in a cold box can be equipped with heating Peltier elements at the ends of maze arms and center, allowing the experimenter to induce ambulation in the one-dimensional arms, followed by extended immobility and sleep at designated areas. In a way, the ThermoMaze is analogous to the water maze (*Morris, 1984*), also an avoidance task, but many more trials can be achieved in a single session and without the inconvenience of a wet environment.

## SPW-R spiking content is biased by current position of the animal and depends on brain state

We demonstrate the utility of the ThermoMaze for addressing long-standing questions in hippocampal physiology. Preparatory and consummatory behaviors in the hippocampus are associated with theta oscillations and SPW-Rs, respectively (*Buzsáki et al., 1983*). SPW-Rs also occur during NREM sleep but studying the differences between waking and sleep SPW-Rs has been hampered by the paucity of SPW-Rs in typical learning paradigms (*Foster and Wilson, 2006*; *Kay et al., 2016*; *Diba and Buzsáki, 2007*; *Dupret et al., 2010*; *Silva et al., 2015*; *Pfeiffer and Foster, 2013*). Neural activity during SPW-Rs has been shown to replay activity patterns observed during previous spatial navigation experiences (*Foster and Wilson, 2006*; *O'Neill et al., 2006*; *Diba and Buzsáki, 2007*) and can even be predictive of activity during future experiences (*Davidson et al., 2009*; *Gupta et al., 2010*; *Karlsson and Frank, 2009*). However, the extent to which SWP-R spiking context is biased by the current position of the animal is less known, as systematic control of position during rest/sleep has posed difficulty. The ThermoMaze enables the experimenter to control the animal's position during SWP-R states. In agreement with previous studies (*O'Neill et al., 2006*; *Dupret et al., 2010*; *Pfeiffer and Foster, 2013*), we found that neurons whose place fields overlapped with the quadrant of the

maze had a higher participation probability in SPW-Rs occurring at that location compared to other neurons. This observation supports the notion that waking replay events can be biased by perceiving features of the surrounding environment (*O'Neill et al., 2006*). However, when the mouse fell asleep at the same location this relationship was weakened but did not disappear. Another potential explanation for the decreased correlation between sleep SPW-R and waking exploration is the deterioration of replay as the function of time (*Wilson and McNaughton, 1994*). Alternatively, the persisting significant correlation between sleep SPW-Rs and previous exploration may also indicate that factors other than the perception of the animal's vicinity are responsible for sleep replay (*Karlsson and Frank, 2009*; *Dragoi and Tonegawa, 2011*; *Grosmark and Buzsáki, 2016*). Continuity of waking experience replay in waking and sleep SPW-Rs have been hypothesized previously but not yet tested (*Jarosiewicz and Skaggs, 2004*). Using the ThermoMaze, this and other related questions can now be addressed quantitatively.

## Materials and methods
### Animals and surgery
All experiments were approved by the Institutional Animal Care and Use Committee at New York University Langone Medical Center (Protocol number: IA15-01466). Animals were handled daily and accommodated to the experimenter and the ThermoMaze before the surgery and electrophysiological recordings. Mice (adult female *n* = 8, 22 g and male *n* = 5, 26 g, C57/Bl6, Strain #:000664, The Jackson Laboratory) were kept in a vivarium on a 12-hr light/dark cycle and were housed two per cage before surgery and individually after it. Atropine (0.05 mg/kg, s.c.) was administered after isoflurane anesthesia induction to reduce saliva production. The body temperature was monitored and kept constant at 36–37°C with a DC temperature controller (TCAT-LV; Physitemp, Clifton, NJ). Stages of anesthesia were maintained by confirming the lack of a nociceptive reflex. The skin of the head was shaved, and the surface of the skull was cleaned by hydrogen peroxide (2%). A custom 3D-printed baseplate (*Vöröslakos et al., 2021a*) (Form2 printer, FormLabs, Somerville, MA) was attached to the skull using C&B Metabond dental cement (Parkell, Edgewood, NY). The location of the craniotomy was marked and a stainless-steel ground screw was placed above the cerebellum. Silicon probe (*Supplementary file 1a*) attached to a metal microdrive (*Vöröslakos et al., 2021b*) was implanted into the dorsal CA1 of the hippocampus (2 mm posterior from Bregma and 1.5 mm lateral to midline) and a copper mesh protective cap was built around the probe. Animals received ketoprofen (5.2 mg/kg, s.c.) at the end of the surgery and on the following 2 days. Each animal recovered at least 5 days prior to experiments. The electrophysiology data were digitized at 20,000 samples/s using an RHD2000 recording system (Intan Technologies, Los Angeles, CA). The number of recorded sessions from each animal is summarized in *Supplementary file 1a*.

### Construction of ThermoMaze
The ThermoMaze is a box (width, length, and height: 20, 20, and 40 cm, respectively), made from acrylic plexiglass sheet (8505K743, McMaster, Elmhurst, IL). The floor of the maze was constructed from 25 Peltier elements (40, 40, and 3.6 mm, Model: TEC1-12706, voltage: 12 V, Umax (V): 15 V, Imax (A): 5.8 A, ΔTmax (Qc = 0): up to 65°C). Each Peltier element was glued inside a custom 3D-printed frame (file can be downloaded from here) using dental cement (Unifast LC, GC America, Alsip, IL) and wood epoxy (Quick-Cure, product number: BSI201, Bob Smith Industries, Atascadero, CA). Once Peltier elements were secured in the 3D-printed frame, an aluminum water-cooling block heatsink (40, 40, and 12 mm; a19112500ux0198, Amazon.com) was attached to each Peltier element using heat-conductive epoxy (8349TFM, MG Chemicals, Ontario, Canada). A variable voltage source (E36102A Power Supply, Keysight Technologies, Santa Rosa, CA) was attached to four Peltier elements using a relay system (4-Channel Relay Module, product number: 101-70-101, SainSmart, Lenexa, KS). The relays were controlled by an Arduino Mega (Arduino Mega 2560 Rev3) running a custom written code. Five aluminum water-cooling block heatsinks were connected together using silicon tubes (5/16″ ID × 7/16″ OD, product number: 5233K59, McMaster, Elmhurst, IL). One of the five heatsinks was connected to a mini submersible electric brushless water pump (240L/H, 3.6W, Ledge, ASIN: B085NQ5VVJ) using silicon tubes and another one was routed to the water tank. We used 5 water pumps to circulate water through the 25 cooling blocks. The water pumps were placed inside a water

tank (40, 40, and 60 cm acrylic box) and were powered using a DC power supply (E3620A, Keysight Technologies, Santa Rosa, CA). The temperature of the water tank was monitored by a K-type thermocouple (5SC-TT-K-40-72, Omega, Norwalk, CT) attached to a handheld thermometer (HH800, Omega, Norwalk, CT) and recorded by a K-type thermocouple (5SC-TT-K-40-72, Omega, Norwalk, CT) attached to an AD595 interface chip (1528-1407-ND, Digi-Key, Thief River Falls, MN) connected to an analog input of the RHD2000 USB Eval system (Intan Technologies, Los Angeles, CA). To monitor the floor temperature of the ThermoMaze, a thermal camera (C5, Flir, Thousand Oaks, CA) was used.

## Behavior

The ThermoMaze setup provides a customized temperature landscape, which the animal can freely explore and choose where to settle. Without any training or shaping, a mouse will search and find the unmarked warm spot and stay on it for extended periods due to thermotaxis (movement toward locations with preferred temperature around 26–29°C; *Figure 3*; *Gaskill et al., 2009*; *Kanosue et al., 1998*). When the heating Peltier element is turned off, the animal quickly leaves the spot and explores the maze again until it finds another warm spot.

On each experimental day, the mouse is taken from the animal facility during their light cycle. The animal is first recorded in its home cage for 1–2 hr (pre-home). It is then transferred into the ThermoMaze under room temperature to freely explore for 10 min (Pre-cooling). During the Pre-cooling sub-session, the water circulation system is circulating room temperature water, and the Peltier elements are not activated. After the Pre-cooling sub-session, 4 kg of ice and two ice packs (25201, Igloo) are added into the water tank while the animal remains in the ThermoMaze. Within 1 min, the temperature of the water in the tank stabilizes at 10–13°C. We then turn on the pump to cool down the ThermoMaze setup (it takes ~120 s to cool down the floor to 10–13°C). At the same time, the Arduino-controlled Peltier element heating system is turned on to heat one of the four $4 \times 4$ cm$^2$ for 5 min, followed by another Peltier device in a fixed sequence (*Figure 2*). Such sequence is repeated four times (total of 80 min) during a Cooling sub-session. After the sub-session, the animal explores again at room temperature for 10 min (Post-cooling sub-session). To increase the temperature back to ~20°C, the ice packs are removed, and 6.5 l of 55°C water is added into the tank. The temperature in the ThermoMaze returns to room temperature within 2 min. After the Post sub-session, recording of electrophysiological activity continues in the home cage for an additional 1–2 hr (post-home cage; *Figure 3A*).

To quantify the behavior of the animal within the ThermoMaze, video is recorded using a Basler camera (a2A2590-60ucBAS Basler ACE2) using the mp4 format with a framerate of 25 Hz. TTL pulses are sent from the camera to the Intan recording system to synchronize the video and the electrophysiological recordings. The animal's location is detected within a $25 \times 25$ cm region of interest (ROI), using a custom-trained DeepLabCut neural network (*Mathis et al., 2018*). Detections with a likelihood below 0.5 are discarded. The occasionally missing trajectory detections are filled using MATLAB function 'fillmissing' with method 'pchip' which is a shape-preserving piecewise cubic spline interpolation and are then smoothed using a seventh-order one-dimensional median filter 'medfilt1'. The detection quality is visually examined by superimposing the detected animal location in each frame on the video.

## Brain temperature measurement

To examine the effects of changing environmental temperature on brain temperature homeostasis, we implanted one male and one female wild-type mice (C57Bl6, 28 g, Strain #:000664, The Jackson Laboratory) with a thermistor (Semitec, 223Fu3122-07U015) in the hippocampus (2 mm posterior from bregma and 1.5 mm lateral to midline) (*Petersen et al., 2022*). After 5 days of postsurgical recovery, the animal was placed inside the ThermoMaze and brain temperature and behavior were monitored ($n$ = 5 sessions, each session consisted of pre-home cage, Pre, Cooling, Post, and post-home cage epochs).

## Quantification and statistical analysis

### SPW-R detection and properties

SPW-Rs were detected as described previously from manually selected channels located in the center of the CA1 pyramidal layer (here). Broadband LFP was bandpass-filtered between 130 and 200 Hz using a third-order Chebyshev filter, and the normalized squared signal was calculated. SPW-R peaks

were detected by thresholding the normalized squared signal at 5× SDs above the mean, and the surrounding SPW-R begin, and end times were identified as crossings of 2×SDs around this peak. SPW-R duration limits were set to be between 20 and 200 ms. An exclusion criterion was provided by manually designating a 'noise' channel (no detectable SPW-Rs in the LFP), and events detected on this channel were interpreted as false positives (e.g., electromyogram [EMG] artifacts). The ripple detection quality was visually examined by superimposing the detected timestamps on the raw LFP traces in NeuroScope2 software suite (*Petersen et al., 2021*).

### Sleep state scoring and LFP analysis

Brain state scoring was performed as described in the study by *Watson et al., 2016*. In short, spectrograms were constructed with a 1-s sliding 10-s window fast Fourier transform of 1250 Hz data at log-spaced frequencies between 1 and 100 Hz. Three types of signals were used to score states: broadband LFP, narrowband high-frequency LFP and EMG calculated from the LFP. For broadband LFP signal, principal component analysis was applied to the Z-transformed (1–100 Hz) spectrogram. The first principal component in all cases was based on power in the low (32 Hz) frequencies. Dominance was taken to be the ratio of the power at 5–10 and 2–16 Hz from the spectrogram. All states were inspected and curated manually, and corrections were made when discrepancies between automated scoring and user assessment occurred.

To quantify the changes in LFP (*Figure 3—figure supplement 3A, B*), we detected wakeful periods in both the home cage and the ThermoMaze environments. Average LFP power and coherence spectra were calculated with Welch's power spectral density method. We used a 4096-point fast Fourier transform, applied to data during wakefulness (non-continuous 2500s in both environments). To compare immobility periods on the ThermoMaze and the home cage, we detected rest epochs that lasted at least 2 s during wakefulness in both environments. This analysis was performed on a subset of animals (*n* = 17 sessions in 7 mice) that had accelerometer signal available (home cage behavior was not monitored by video). Using these behavior transition timepoints, we calculated the event triggered power spectra for delta and theta bands (±2 s around the transition time).

### Unit isolation and classification

A concatenated signal file was prepared by merging all recordings from a single animal from a single day. Putative single units were first sorted using Kilosort (*Pachitariu et al., 2016*) and then manually curated using Phy (https://phy-contrib.readthedocs.io/). After extracting timestamps of each putative single unit activity, the spatial tuning properties, identification of 2D place cells and place fields, and participation in SPW-Rs events were analyzed using customized MATLAB (Mathworks, Natick, MA) scripts.

In the processing pipeline, cells were classified into three putative cell types: narrow interneurons, wide interneurons, and pyramidal cells. Interneurons were selected by two separate criteria; narrow interneurons were assigned if the waveform trough-to-peak latency was less than 0.425 ms. Wide interneuron was assigned if the waveform trough-to-peak latency was more than 0.425 ms and the rise time of the autocorrelation histogram was more than 6 ms. The remaining cells were assigned as pyramidal cells (*Petersen et al., 2021*). We have isolated 1438 putative single units from 7 animals in 20 sessions (*n* = 1150 putative pyramidal cells, *n* = 288 putative interneurons) during the ThermoMaze behavior. We also collected 228 putative pyramidal cells from 2 animals in 3 control sessions (*Figure 3—figure supplement 4*) and 434 putative single units from 4 mice in 7 sessions using the 20-min warmth paradigm (*Figure 6*).

### Pyramidal cells firing and SPW-R rate maps

To visualize and compare the spatial tuning properties of neurons across sub-sessions (Pre, Cooling, and Post) during movement (speed ≥2.5 cm/s), we first binned the ThermoMaze ROI into 25 by 25 bins (each with size 1 × 1 cm) and counted the number of spikes of a neuron that occurred in each bin when the animal was actively moving (movement spike-count map). Next, we summed the total duration of time (in seconds) that the animal spent moving in each spatial bin to construct the 'movement occupancy map'. The sub-session rate map of a cell during movement was computed by dividing the spike-count map by the occupancy map bin-wise. Similarly, we computed the SPW-R rate map within a sub-session by dividing the number of ripples that occurred in each bin by the total duration of

immobility (speed <2.5 cm/s) that the animal spent in each bin. Both firing and SPW-R rate maps were spatially smoothed using a 2-bin smoothing window (see here).

## Spatial tuning of spikes during SPW-Rs

To quantify the spatial tuning of neurons during SPW-Rs (*Figure 5*), we defined a metric called 'within-ripple spatial tuning score' which is a value between 0 and 1. The higher score indicates stronger spatial tuning of a neuron during SPW-Rs. We first binned the ThermoMaze ROI into four quadrants (2 × 2) and determined the firing rate of the neuron in each quadrant within SPW-Rs (i.e., total number of spikes of the cell divided by the total duration of SPW-R in that quadrant). For each SPW-R, a 300-ms time window surrounding the ripple's power peak time was taken and the temporal overlaps between SPW-Rs were removed. Next, the within-SPW-R firing rate ratio in a given quadrant (e.g., in quadrant A), is defined to be the firing rate of the neuron during SPW-Rs in quadrant A divided by the sum of the within-SPW-R firing rate in all four quadrants. Finally, the within-ripple STS (*Figure 5*) of a neuron is defined to be the maximum within-SPW-R firing rate ratio of the cell among all quadrants. To test the hypothesis that such spatial tuning exists beyond chance level, we generated shuffled within-SPW-R firing rate maps by randomly assigning one of the four quadrants to each SPW-R. Specifically, we randomly permuted the location of the SPW-Rs so that the number of SPW-Rs per quadrant was kept fixed for the shuffled condition.

## Bayesian decoding of the animal position

Bayesian decoding of the animal's position was based on the method provided by *Zhang et al., 1998*. In short, we utilized the spatial firing rate maps constructed to find the location that maximally explains the observation of spiking within a certain time window. Because SPW-Rs occurred mainly in the corners where the warm spots were, we simplified the analysis and binned the ThermoMaze into 2 × 2 quadrants, which yielded four maze areas. We constructed the firing rate map templates $f_i(x)$ of each neuron during SPW-Rs (300-ms time window surrounding the peak of each SPW-R) within the Cooling sub-session. The decoded position was then determined to be the quadrant that maximizes the posterior likelihood given the observed spike counts:

$$P\left(\boldsymbol{x}|\boldsymbol{n}\right) = C\left(\tau,\boldsymbol{n}\right) P\left(x\right) \left(\prod_{i=1}^{N} f_i\left(\boldsymbol{x}\right)^{n_i}\right) exp\left(-r\sum_{i=1}^{N} f_i\left(\boldsymbol{x}\right)\right)$$

where $x$ was the quadrant index, $n$ was the spike-count vector observed surrounding the frame time, $\tau$ was the time window size and equals 300 ms, $C(\tau,n)$ was a normalization factor and was taken to be 1, $P(x)$ was the prior probability distribution of animal location and was taken to be 1 in the case of *Figure 5D*, $i$ was the index of each cell, $f_i(x)$ was the average firing rate of cell $i$ at position **x**, and $N$ was the total number of pyramidal cells recorded in the session. For the purpose of cross-validation, we divided the SPW-Rs in each session into 100 folds. For each fold (testing dataset), the firing rate map templates were constructed using SPW-Rs from the other 99 folds (training dataset), and the decoding accuracy for the omitted fold was computed as the proportion of SPW-Rs whose corresponding quadrant was correctly decoded over the total number of SPW-Rs in the fold. For each session, we report the average decoding accuracy of test datasets.

## Comparison of spatial tuning during SPW-Rs and movement

To quantify the similarity between spatial tuning of neurons during SPW-R and movement (theta oscillation), we calculated the firing rate ratios during movement in a similar way as we calculated the within-SPW-R firing rate ratios (see section 'Spatial tuning during SPW-Rs'). The ThermoMaze ROI was again binned into quadrants and firing rate maps (2 × 2) of each neuron during movement were calculated. The firing rate ratio of a neuron in each quadrant during movement was defined as the quadrant with the actual firing rate in that quadrant divided by its mean firing rate in all quadrants. Next, the Pearson correlation between the firing rate ratios during SPW-Rs and movement in each quadrant for each cell within the Cooling sub-sessions were calculated.

We also studied the correlation between pyramidal cells' spatial tuning during SPW-Rs and movement at a population level (*Figure 6G*). In each session, we first constructed population vectors in each quadrant by concatenating the firing rate ratio of each cell in a quadrant into a vector during

SPW-R or movement. We then computed the pairwise correlation coefficients among the four population vectors between each condition in the correlation matrix and took the average across sessions.

## Place cells identification

Data recorded in the ThermoMaze were used for analyzing the spatial tuning of spiking activity. Each session was split into three sub-sessions and all the criteria described below were independently applied to each sub-session. Putative pyramidal units with peak firing rates lower than 0.4 Hz and spatial information content (*Mizuseki et al., 2012*; *Skaggs et al., 1992*) lower than 0.25 bits/spike were not considered place cells (*Roux et al., 2017*). We further computed the chance-level spatial information distribution by generating 100 shuffle datasets (by parsing spike trains and the mouse position during running into 5 s blocks and randomly shuffling the temporal correspondence between the spiking and position bins). If the *z*-scored spatial information (according to the chance-level spatial information distribution) is smaller than 1.65 (p-value >0.05 in a one-sided test), the cell was not considered to be a place cell (*Roux et al., 2017*; *Markus et al., 1994*). In the end, a pyramidal neuron is determined to be a place cell throughout the entire session if it meets the criteria in all three individual sub-sessions.

## Data and code availability

Matlab codes used for analyzing the data are available at https://github.com/buzsakilab/buzcode (copy archived at *Levenstein et al., 2025*). The dataset is available at https://zenodo.org/records/11235921.

## Acknowledgements

We thank Daniel Levenstein for useful comments on the manuscript. We thank Yiyao Zhang, Anna Maslarova, and Leeor Alon for their help with different aspects related to the experiments. Supported by MH122391 and U19NS107616.

## Additional information

### Funding

| Funder | Grant reference number | Author |
|---|---|---|
| National Institutes of Health | U19NS107616 | György Buzsáki |
| National Institutes of Health | MH122391 | György Buzsáki |

The funders had no role in study design, data collection and interpretation, or the decision to submit the work for publication.

### Author contributions

Mihály Vöröslakos, Conceptualization, Data curation, Formal analysis, Writing – original draft, Writing – review and editing; Yunchang Zhang, Data curation, Formal analysis, Writing – original draft, Writing – review and editing; Kathryn McClain, Roman Huszár, Formal analysis, Writing – original draft; Aryeh Rothstein, Data curation, Writing – original draft; György Buzsáki, Supervision, Writing – original draft, Writing – review and editing

### Author ORCIDs

Mihály Vöröslakos ⬡ https://orcid.org/0000-0002-1022-1355
Yunchang Zhang ⬡ http://orcid.org/0000-0003-3294-7373
György Buzsáki ⬡ https://orcid.org/0000-0002-3100-4800

### Ethics

All experiments were approved by the Institutional Animal Care and Use Committee at New York University Langone Medical Center (Protocol number: IA15-01466).

Reviewer #1 (Public Review): https://doi.org/10.7554/eLife.90347.3.sa1
Author response https://doi.org/10.7554/eLife.90347.3.sa2

## Additional files

### Supplementary files

Supplementary file 1. Animal sujects and statistical information. (**a**) Summary of animal subjects with brain implants. (**b**) p-values of multiple group comparisons pertaining to analyses of variance in *Figure 3C*. Cumulative distribution of animal speed in the ThermoMaze during three sub-sessions. (**c**) p-values of multiple group comparisons pertaining to analyses of variance in *Figure 4*. Box plots of Pearson correlation coefficients between spatial firing rate maps. Here, group numbers 1, 2, 3, and 4 refer to correlation values between Pre and Cooling, Cooing and Post, and Pre and Post in control sessions. (**d**) p-values of multiple group comparisons pertaining to analyses of variance in *Figure 5—figure supplement 1A*. Pyramidal neurons increase firing rate during ripples in their preferred quadrant during movement. Numbers 1–8 represent pyramidal firing rate: 1. during SPW-R inside the cell's preferred quadrant during ripple, 2. during SPW-R outside the cell's preferred quadrant during ripple, 3. during SPW-R inside the cell's preferred quadrant during movement, 4. during SPW-R outside the cell's preferred quadrant during movement, 5. during SPW-R inside the cell's preferred quadrant during ripple, 6. during SPW-R outside the cell's preferred quadrant during ripple, 7. during SPW-R inside the cell's preferred quadrant during movement, and 8. during SPW-R outside the cell's preferred quadrant during movement. (**e**) p-values of multiple group comparisons pertaining to analyses of variance in *Figure 5—figure supplement 1B*. Interneurons firing rate does not change during ripples in their preferred quadrant during movement. Numbers 1–8 represent interneuron firing rate: 1. during ripples inside the cell's preferred quadrant during ripple, 2. during ripples outside the cell's preferred quadrant during ripple, 3. during ripples inside the cell's preferred quadrant during movement, 4. during non-ripples outside the cell's preferred quadrant during movement, 5. during non-ripples inside the cell's preferred quadrant during ripple, 6. during non-ripples outside the cell's preferred quadrant during ripple, 7. during non-ripples inside the cell's preferred quadrant during movement, and 8. during non-ripples outside the cell's preferred quadrant during movement.

MDAR checklist

### Data availability

Spiking, local field potential events, and behavioral data have been deposited in Zenodo (https://doi.org/10.5281/zenodo.11235921).

The following dataset was generated:

| Author(s) | Year | Dataset title | Dataset URL | Database and Identifier |
|---|---|---|---|---|
| Voroslakos M | 2024 | ThermoMaze: A behavioral paradigm for readout of immobility-related brain events | https://doi.org/10.5281/zenodo.11235921 | Zenodo, 10.5281/zenodo.11235921 |

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
