## [Editor Report · eLife assessment]

The ThermoMaze represents a **valuable** tool to control the rest/exploration states of an animal. The data, collected and analyzed using **solid** and validated methodology, demonstrate its use in addressing previously elusive questions. This will facilitate future work with more in-depth analysis of place cell activity to further support for some of the claims.

---

## [Referee Report · Reviewer #1 (Public Review)]

Summary:

This manuscript introduced a new behavioral apparatus to regulate the animal's behavioral state naturally. It is a thermal maze where different sectors of the maze can be set to different temperatures; once the rest area of the animal is cooled down, it will start searching for a warmer alternative region to settle down again. They recorded with silicon probes from the hippocampus in the maze and found that the incidence of SWRs was higher at the rest areas and place cells representing a rest area were preferentially active during rest-SWRs as well but not during non-REM sleep.

Strengths:

The maze can have many future applications, e.g., see how the duration of waking immobility can influence learning, future memory recall, or sleep reactivation. It represents an out-of-the-box thinking to study and control less-studies aspects of the animals' behavior.

Weaknesses:

The impact is only within behavioral research and hippocampal electrophysiology.

---

## [Author Response]

The following is the authors’ response to the original reviews.

**Reviewer #1 (Public Review):**
This manuscript introduced a new behavioral apparatus to regulate the animal's behavioral state naturally. It is a thermal maze where different sectors of the maze can be set to different temperatures; once the rest area of the animal is cooled down, it will start searching for a warmer alternative region to settle down again. They recorded with silicon probes from the hippocampus in the maze and found that the incidence of SWRs was higher at the rest areas and place cells representing a rest area were preferentially active during rest-SWRs as well but not during non-REM sleep.

We thank the reviewer for carefully reading our manuscript and providing useful and constructive comments.

Strengths:The maze can have many future applications, e.g., see how the duration of waking immobility can influence learning, future memory recall, or sleep reactivation. It represents an out-of-the-box thinking to study and control less-studies aspects of the animals' behavior.Weaknesses:The impact is only within behavioral research and hippocampal electrophysiology.

We agree with this assessment but would like to add that the intersection of electrophysiological recordings in behaving animals is a very large field. Behavioral thermoregulation is a hotly researched area also by investigators using molecular tools as well. The ThermoMaze can be used for juxtacellular/intracellular recordings in behaving animals. Restricting the animal’s movement during these recordings can improve the length of recording time and recorded single unit yield in these experiments.

Moreover, the fact that animals can sleep within the task can open up new possibilities to compare the role of sleep in learning without having to move the animal from a maze back into its home cage. The cooling procedure can be easily adapted to head-fixed virtual reality experiments as well.

I have only a few questions and suggestions for future analysis if data is available.Comment-1: Could you observe a relationship between the duration of immobility and the preferred SWR activation of place cells coding for the current (SWR) location of the animal? In the cited O'Neill et al. paper, they found that the 'spatial selectivity' of SWR activity gradually diminished within a 2-5min period, and after about 5min, SWR activity was no longer influenced by the current location of the animal. Of course, I can imagine that overall, animals are more alert here, so even over more extended immobility periods, SWRs may recruit place cells coding for the current location of the animal.

We thank the reviewer for raising this question, which is a fundamental issue that we attempted to address using the ThermoMaze. First, we indeed observed persistent place-specific firing of CA1 neurons for up to around 5 minutes, which was the maximal duration of each warm spot epoch, as shown by the decoding analysis (based on firing rate map templates constructed during SPW-Rs) in Figure 5C and D. However, we did not observe above-chance-level decoding of the current position of the animal during sharp-wave ripples using templates constructed during theta, which aligns with previous observation that CA1 neurons during “iSWRs” (15–30 s time windows surrounding theta oscillations) did not show significant differences in their peak firing rate inside versus outside the place field (O’Neil et al., 2006). We reasoned that this could be potentially explained by a different (although correlated, see Figure 5E) neuronal representation of space during theta and during awake SPW-R.

Comment-2: Following the logic above, if possible, it would be interesting to compare immobility periods on the thermal maze and the home cage beyond SWRs, as it could give further insights into differences in rest states associated with different alertness levels. E.g., power spectra may show a stronger theta band or reduced delta band compared to the home cage.

If we are correct the Reviewer would like to know whether the brain state of the animal was similar in the ThermoMaze (warm spot location) and in the home cage during immobility. A comparison of the time-evolved power spectra shows similar changes from walking to immobility in both situations without notable differences. This analysis was performed on a subset of animals (n = 17 sessions in 7 mice) that were equipped with an accelerometer (home cage behavior was not monitored by video). We detected rest epochs that lasted at least 2 seconds during wakefulness in both the home cage and ThermoMaze. Using these time points we calculated the event-triggered power spectra for the delta and theta band (±2 s around the transition time) and found no difference between the home cage and ThermoMaze (Suppl. Fig. 4D).

Prompted by the Reviewer’s question, we further quantified the changes in LFP in the two environments. We did not find any significant change in the frequencies between 1-40 Hz during Awake periods, but we did find higher delta power (1-4 Hz) in some animals in the ThermoMaze (Suppl. Fig. 4A, B).

We have also quantified the delta and theta power spectra in the few cases, when the warm spot was maintained, and the animal fell asleep. The time-resolved spectra classified the brain state as NREM, similar to sleeping in the home cage. Both delta and theta power were higher in the ThermoMaze following Awake-NREM transitions (±30 seconds around the transition, Suppl. Fig. 4C). It might well be that immobility/sleep outside the mouse’s nest might reflect some minor (but important) differences but our experiments with only a single camera recording do not have the needed resolution to reveal minor differences in posture.

We added these results to the revised Supplementary material (Suppl. Fig. 4).

Comment-3: Was there any behavioral tracking performed on naïve animals that were placed the first time in the thermal maze? I would expect some degree of learning to take place as the animal realizes that it can find another warm zone and that it is worth settling down in that area for a while. Perhaps such a learning effect could be quantified.

Unfortunately, we did not record videos during the first few sessions in the ThermoMaze. Typically, we transferred a naïve animal into the ThermoMaze for an hour on the first day to acclimatize them to the environment. This was performed without video analysis. In addition, because the current version of the maze is relatively small (20 x 20 cm), the animal usually walked around the edges of the maze before settling down at a heated warm spot. It appeared to us that there was only a very weak drive to learn the sequence and location of the warm spot, and therefore we did not quantified learning in the current experiment. We agree with the reviewer that in future studies, it will be interesting to explore whether the ThermoMaze could be adapted to a land-version of the Morris water maze by increasing the size of the maze and performing more controlled behavioral training and testing.

Comment-4: There may be a mislabeling in Figure 6g because the figure does not agree with the result text - the figure compares the population vector similarly of waking SWR vs sleep SWRs to exploration vs waking SWR and exploration vs sleep SWRs.

We thank the reviewer for raising the point, we have updated the labels accordingly.

**Reviewer #2 (Public Review):**
In this manuscript, Vöröslakos and colleagues describe a new behavioural testing apparatus called ThermoMaze, which should facilitate controlling when a mouse is exploring the environment vs. remaining immobile. The floor of the apparatus is tiled with 25 plates, which can be individually heated, whereas the rest of the environment is cooled. The mouse avoids cooled areas and stays immobile on a heated tile. The authors systematically changed the location of the heated tile to trigger the mouse's exploratory behaviours. The authors showed that if the same plate stays heated longer, the mouse falls into an NREM sleep state. The authors conclude their apparatus allows easy control of triggering behaviours such as running/exploration, immobility and NREM sleep. The authors also carried out single-unit recordings of CA1 hippocampal cells using various silicone probes. They show that the location of a mouse can be decoded with above-chance accuracy from cell activity during sharp wave ripples, which tend to occur when the mouse is immobile or asleep. The authors suggest that consistent with some previous results, SPW-Rs encode the mouse's current location and any other information they may encode (such as past and future locations, usually associated with them).

We thank the reviewer for carefully reading our manuscript and providing useful and constructive comments.

Strengths:Overall, the apparatus may open fruitful avenues for future research to uncover the physiology of transitions from different behavioural states such as locomotion, immobility, and sleep. The setup is compatible with neural recordings. No training is required.Weaknesses:I have a few concerns related to the authors' methodology and some limitations of the apparatus's current form. Although the authors suggest that switching between the plates forces animal behaviour into an exploratory mode, leading to a better sampling of the enclosure, their example position heat maps and trajectories suggest that the behaviour is still very stereotypical, restricted mostly to the trajectories along the walls or the diagonal ones (between two opposite corners). This may not be ideal for studying spatial responses known to be affected by the stereotypicity of the animal's trajectories. Moreover, given such stereotypicity of the trajectories mice take before and after reaching a specific plate, it may be that the stable activity of SWR-P ripples used for decoding different quadrants may be representing future and/or past trajectories rather than the current locations suggested by the authors. If this is the case, it may be confusing/misleading to call such activity ' place-selective firing', since they don't necessarily encode a given place per se (line 281).

We agree with the reviewer that the current version of the ThermoMaze does not necessarily motivate the mice to sample the entire maze during warm spot transitions. However, we did show correlational evidence that neuronal firing during awake sharp-wave ripples is place-selective. Both firing rate ratios and population vectors of CA1 neurons showed a reliable correlation between those during movement and awake sharp-wave ripples (Figure 5 E and F), indicating that spatial coding during movement persists into awake SWR-P state. This finding rejects the hypothesis that neuronal firing during ripples throughout the Cooling sub-session encodes past/future trajectories, which could be explained by a lack of goal-directed behavior in order to perform the task. We hope to test whether such place-specific firing during ripples can be causally involved in maintaining an egocentric representation of space in a future study.

Besides, we have attempted to motivate the animal to visit the center of the maze during the Cooling sub-session. Moving the location of warm spots from the corners can shape the animals’ behavior and promote more exploration of the environment as we show in Suppl. Fig. 5. We agree with the Reviewer that the current size of the ThermoMaze poses these limitations. However, an example future application could be to warm the floor of a radial-arm maze by heating Peltier elements at the ends of maze arms and center in an otherwise cold room, allowing the experimenter to induce ambulation in the 1-dimensional arms, followed by extended immobility and sleep at designated areas.

Another main study limitation is the reported instability of the location cells in the Thermomaze. This may be related to the heating procedure, differences in stereotypical sampling of the enclosure, or the enclosure size (too small to properly reveal the place code). It would be helpful if the authors separate pyramidal cells into place and non-place cells to better understand how stable place cell activity is. This information may also help to disambiguate the SPW-R-related limitations outlined above and may help to solve the poor decoding problem reported by the authors (lines 218-221).

The ThermoMaze is a relatively small enclosure (20 x 20 cm) compared to typical 2D arenas (60 x 60 cm) used in hippocampal spatial studies. Due to the small environment, one possibility is that CA1 neurons encode less spatial information and only a small number of place cells could be found. Therefore, we identified place cells in each sub-session. We found 40.90%, 45.32%, and 41.26% of pyramidal cells to be place cells in the Pre-cooling, Cooling, and Post-cooling sub-sessions, respectively. Furthermore, we found on average 17.36% of pyramidal neurons pass the place cell criteria in all three sub-sessions in a daily session. Therefore, the strong decorrelation of spatial firing maps across sub-sessions cannot be explained by poor recording quality or weak neuronal encoding of spatial information but is potentially due to changes in environmental conditions.

Some additional points/queries:Comment-1: Since the authors managed to induce sleeping on the warm pads during the prolonged stays, can they check their hypothesis that the difference in the mean ripple peak frequency (Fig. 4D) between the home cage and Thermomaze was due to the sleep vs. non-sleep states?

In response to the reviewer’s comment, we compared the ripple peak frequency that occurred during wakefulness and NREM epochs in the home cage and ThermoMaze (n = 7 sessions in 4 mice). We found that the peak frequency of the awake ripples was higher compared to both home cage and ThermoMaze NREM sleep (one-way ANOVA with Tukey’s posthoc test, ripple frequencies were: 171.63 ± 11.69, 172.21 ± 11.86, 168.19 ± 11.10 and 168.26 ± 11.08 Hz mean ± SD for home cage awake, ThermoMaze awake, home cage NREM and ThermoMaze NREM conditions, p < 0.001 between awake and NREM states). We added this quantification to the revised manuscript.

**Author response image 1. sa2fig1:** NREM sleep either in home cage or in ThermoMaze affects ripple mean peak frequency similarly.

Comment-2: How many cells per mouse were recorded? How many of them were place cells? How many place cells at the same time on average? What are the place field size, peak, and mean firing rate distributions in these various conditions? It would be helpful if they could report this.

For each animal on a given day, the average number of cells recorded was 57.5, which depended on the electrodes and duration after implantation. We first applied peak firing rate and spatial information thresholds to identify place cells in each sub-session (see more details in the revised Methods section for place cell definition). We found 40.90%, 45.32%, and 41.26% of pyramidal cells to be place cells in the Pre-cooling, Cooling, and Post-cooling sub-sessions respectively. Furthermore, we found on average 17.36% of pyramidal neurons pass the place cell criteria in all three sub-sessions in a daily session.

For place cells identified in each sub-session, their place fields size is on average 61.03, 79.86, and 57.51 cm2 (standard deviation = 60.13, 69.98, and 49.64 cm2; Pre-cooling, Cooling, and Post-cooling correspondingly). A place field was defined to be a contiguous region of at least 20 cm2 (20 spatial bins) in which the firing rate was above 60% of the peak firing rate of the cell in the maze (Roux and Buzsaki et al., 2017). A place field also needs to contain at least one bin above 80% of the peak firing rate in the maze. With such definition, the average place field peak firing rate is 5.84, 5.22, and 6.48 Hz (standard deviation = 5.11, 4.65, and 5.83 Hz) and the average mean firing rate within the place fields is 4.54, 4.05, and 5.07 Hz (standard deviation = 4.00, 3.60, and 4.60).

We would like to point out that these values depend strongly on the definition of place fields, which vary widely across studies. We reason that the ThermoMaze paradigm induced place field remapping which has been reported to occur upon changes in the environment such as visual cues (Leutgeb et al., 2009). We hypothesize that temperature gradient is an important aspect among the environmental cues, thus remapping is expected. Overall, we did not aim for biological discoveries in the first presentation of the ThermoMaze. Instead, our limited goal was the detailed description of the method and its validation for behavioral and physiological experiments.

References

(1) Mizuseki K, Royer S, Diba K, Buzsáki G. Activity dynamics and behavioral correlates of CA3 and CA1 hippocampal pyramidal neurons. Hippocampus. 2012 Aug;22(8):1659-80. doi: 10.1002/hipo.22002. Epub 2012 Feb 27. PMID: 22367959; PMCID: PMC3718552.

(2) Skaggs WE,McNaughton BL,Gothard KM,Markus EJ. 1993. An information-theoretic approach to deciphering the hippocampal code. In: SJ Hanson, JD Cowan, CL Giles, editors. Advances in Neural Information Processing Systems, Vol. 5. San Francisco, CA: Morgan Kaufmann. pp 1030–1037.

(3) Roux L, Hu B, Eichler R, Stark E, Buzsáki G. Sharp wave ripples during learning stabilize the hippocampal spatial map. Nat Neurosci. 2017 Jun;20(6):845-853. doi: 10.1038/nn.4543. Epub 2017 Apr 10. PMID: 28394323; PMCID: PMC5446786.

(4) Markus, E.J., Barnes, C.A., McNaughton, B.L., Gladden, V.L. & Skaggs, W.E. Spatial information content and reliability of hippocampal CA1 neurons: effects of visual input. Hippocampus 4, 410–421 (1994).